# Pan-Drug Resistant *Acinetobacter baumannii*, but Not Other Strains, Are Resistant to the Bee Venom Peptide Melittin

**DOI:** 10.3390/antibiotics9040178

**Published:** 2020-04-14

**Authors:** Karyne Rangel, Guilherme Curty Lechuga, André Luis Almeida Souza, João Pedro Rangel da Silva Carvalho, Maria Helena Simões Villas Bôas, Salvatore Giovanni De Simone

**Affiliations:** 1FIOCRUZ, Center for Technological, Development in Health (CDTS)/National, Institute of Science and Technology for Innovation in Neglected Population Diseases (INCT-IDPN), Rio de Janeiro 21040-900; guilherme.lechuga@yahoo.com.br (G.C.L.); alsouza@ioc.fiocruz.br (A.L.A.S.); joaopedrorsc@gmail.com (J.P.R.d.S.C.); 2FIOCRUZ, Oswaldo Cruz Institute, Laboratory of Cellular Ultrastructure, Rio de Janeiro 21040-900, Brazil; 3FIOCRUZ, Microbiology Department, National Institute for Quality Control in Health (INCQS), Rio de Janeiro 21040-900, Brazil; maria.villas@incqs.fiocruz.br; 4FIOCRUZ, Federal Fluminense University, Biology Institute, Department of Molecular and Cellular Biology, Rio de Janeiro, Niterói 24020-140, Brazil

**Keywords:** *Acinetobacter baumannii*, multidrug resistance, pan drug resistance, melittin, biofilm, antimicrobial peptide

## Abstract

*Acinetobacter baumannii* is a prevalent pathogen in hospital settings with increasing importance in infections associated with biofilm production. Due to a rapid increase in its drug resistance and the failure of commonly available antibiotics to treat *A. baumannii* infections, this bacterium has become a critical public health issue. For these multi-drug resistant *A. baumannii*, polymyxin antibiotics are considered the only option for the treatment of severe infections. Concerning, several polymyxin-resistant *A. baumannii* strains have been isolated over the last few years. This study utilized pan drug-resistant (PDR) strains of *A. baumannii* isolated in Brazil, along with susceptible (S) and extreme drug-resistant (XDR) strains in order to evaluate the in vitro activity of melittin, an antimicrobial peptide, in comparison to polymyxin and another antibiotic, imipenem. From a broth microdilution method, the determined minimum inhibitory concentration showed that S and XDR strains were susceptible to melittin. In contrast, PDR *A. baumannii* was resistant to all treatments. Treatment with the peptide was also observed to inhibit biofilm formation of a susceptible strain and appeared to cause permanent membrane damage. A subpopulation of PDR showed membrane damage, however, it was not sufficient to stop bacterial growth, suggesting that alterations involved with antibiotic resistance could also influence melittin resistance. Presumably, mutations in the PDR that have arisen to confer resistance to widely used therapeutics also confer resistance to melittin. Our results demonstrate the potential of melittin to be used in the control of bacterial infections and suggest that antimicrobial peptides can serve as the basis for the development of new treatments.

## 1. Introduction

The emergence of drug-resistant strains of major pathogenic bacteria is an increasingly critical public health issue [1,2]. One such pathogen is the Gram-negative bacterium *Acinetobacter baumannii*, which possesses both multidrug resistance (MDR) genes and an intrinsic natural resistance toward many currently available antibiotics that can lead to untreatable infections [3,4]. The Centers for Disease Control and Prevention (CDC) categorized MDR *Acinetobacter* as a serious threat requiring continuous public health monitoring and prevention activities [5]. To date, *A. baumannii* has shown resistance to a large spectrum of cephalosporins, aminoglycosides, fluoroquinolones and carbapenems as well as polymyxins, a last-resort class of antibiotics [6,7,8]. In 2017, the World Health Organization (WHO) released a priority list of pathogens threatening human health and classified *A. baumannii* as the top critical pathogen for research and development of new antimicrobials [9].

Another critical issue for treatment of *A. baumannii* is its ability to form biofilms, the structured surface-associated multicellular communities of microbes that are encased in an extracellular matrix consisting of proteins, lipids, nucleic acids and polysaccharides [4]. The formation of a biofilm can be a major contributor to treatment failures as its structure can create a physical barrier that prevents antibiotic delivery [10,11,12,13]. In addition to providing greater protection against antibiotics, biofilms can also interfere with host immune defenses and isolate bacteria from adverse environmental conditions [14]. It is estimated that 65–80% of human infections are caused by biofilm-forming bacteria [15], which is responsible for considerable morbidity and contribute significantly to healthcare costs [16,17,18]. Regarding *A. baumannii*, it can form biofilm communities on most abiotic surfaces in hospitals, such as stainless steel and polycarbonate, that can lead to the contamination of equipment, prosthetics, endotracheal tubes, and catheters, as well as biotic surfaces of patients including skin, lung, heart, bladder and other organs [19].

Therapeutic options for multidrug-resistant (MDR) strains of *A. baumannii* infections are limited. One promising source of novel treatments is antimicrobial peptides (AMPs), which have gained increased attention as one of the main options to overcome antibiotic resistance [20,21]. AMPs are typically short peptides (i.e., 2–100 amino acids) that incorporate mostly cationic, hydrophobic and amphipathic properties [22,23,24]. In nature, they are a component of the first line of defense against invading microbes in the host immune defense system of all classes of organisms including microorganisms, plants, insects, fish, reptiles, and mammals [22,23]. The interest in applying AMPs as a therapeutic alternative against pathogenic microorganisms is related to their potency, rapid action and display of a broad spectrum of activities against both gram-negative and gram-positive bacteria as well as viruses, fungi and parasites [21,25,26]. The main mechanism of action attributed to AMPs against pathogens is to target microbial membranes for disruption, destabilization or permeabilization via their formation of pores [23]. Additional mechanisms have been described related to the intracellular translocation of the peptide that can inhibit macromolecular synthesis including DNA, RNA and proteins [27].

One of the most extensively studied AMPs is a major component in the venom of European honeybee *Apis mellifera*, melittin [28]. Melittin has demonstrated a wide range of bactericidal activity against both reference and clinical strains [28,29,30], which includes antibiotic-resistant bacteria, such as *A. baumannii* and *Pseudomonas aeruginosa* [31,32,33]. It is a small cationic linear peptide composed of 26 amino acid residues (GIGAVLKVLTTGLPALISWIKRKRQQ-CONH_2_) with a net charge at physiological pH of +6 due to the presence of arginine and lysine residues [22,34]. The N- and C-terminal amino acids of melittin are mostly hydrophobic and hydrophilic, respectively [28,35]. Polar and nonpolar amino acid residues are distributed asymmetrically in melittin, suggesting an amphipathic nature when it adopts in an α-helical conformation [36]. This cationic and amphipathic structure is regarded as the most characteristic configuration of AMPs, which makes melittin a representative model peptide for understanding the mechanisms of membrane permeabilization by AMPs [28,37,38].

Melittin has also been shown to exert a problematic allergy-based activity by increasing serum immunoglobulin E (IgE) in nearly one-third of honeybee venom-sensitive individuals [39]. Besides, melittin can be incorporated into the phospholipid bilayers of cell membranes that induce dose and time-dependent morphological changes leading to cell lysis. Thus, possible adverse effects of melittin should be considered before evaluating their possible therapeutic applications. Recently, a study demonstrated that topical administration of melittin at concentrations of 16 and 32 µg/mL in mice killed 93.3% and 100% of an XDR *A. baumannii* on a third-degree burned area, respectively [32]. No toxicity was observed on injured or healthy derma, as well as circulating red blood cells in the examined mice. This finding has encouraged further investigations to re-examine the application of naturally occurring AMPs for at least topical treatments, which have been understudied due to the potential toxicity against mammalian cells.

Here, we report on an evaluation of the in vitro activity of melittin against multiple strains collected in Brazilian hospitals that could be described as susceptible (S), extreme drug-resistant (XDR), and pan drug-resistant (PDR). The peptide was also tested for effectiveness against biofilms and its membrane lysing properties by fluorescence microscopy.

## 2. Materials and Methods

### 2.1. Bacterial Strains

Four *A. baumannii* strains were collected from two public hospitals in Rio de Janeiro: one (31852) that was susceptible to eleven antimicrobials tested of six groups, two (33677 and 96734) that harbor *bla*_OXA−23_ genes representing the two major clusters of XDR *A. baumannii* disseminated in Brazil— ST15/CC15 (https://pubmlst.org/bigsdb?page=info&db=pubmlst_abaumannii_isolates&id=3655) and ST79/CC79 (https://pubmlst.org/bigsdb?page=info&db=pubmlst abaumannii_isolates &id= 3647)—according to the Pasteur Institute and one (100) PDR strain that displayed resistance to all antimicrobials tested, including polymyxin. As a reference, the ATCC strain 19606 was included in all antibacterial assays.

### 2.2. Peptide

The peptide melittin used in this study was obtained from Sigma-Aldrich (St. Louis, MO, USA). The peptide was dissolved in distilled water and the solution was stored at −20 °C.

### 2.3. Determination of the Minimum Inhibitory Concentration (MIC)

The MICs for melittin, polymyxin and imipenem were determined by the broth microdilution assays according to the recommendations of the Clinical and Laboratory Standards Institute (CLSI) [40]. After growth at 37 °C for 24 h in nutrient agar medium, bacterial strains were suspended in sterile saline (0.85%) to 0.5 McFarland standard (1 × 10^8^ CFU/mL) and then diluted in the range of 1:100 in cation-adjusted Mueller Hinton (CAMH) broth (pH 7.3) to a final concentration of 1 × 10^6^ CFU/mL. Serial dilutions of the melittin and antibiotics were prepared in culture medium at a volume of 100 µL in 96-well plates. The quantity of melittin ranged from 14 to 85 µg/mL and antibiotics from 0.125 to 64 µg/mL. Their addition to the inoculated plates was executed in less than 30 min followed by mixing on the bench in rotational movement. After a 24 h incubation at 37 °C, bacterial growth was detected by the addition of resazurin to 0.02% and a 1 h incubation [41]. The lowest quantity of antibacterial agents producing complete inhibition of visible growth was considered as the MIC. Controls were included for sterility and bacterial growth along with the reference strains *Escherichia coli* (ATCC 25922) and *P. aeruginosa* (ATCC 27853) as the quality controls for the anti-microbial agents [40].

### 2.4. Inhibition of Biofilm Formation

Biofilm formation by all experimental strains was identified and quantified using a tissue culture plate (TCP) method as described previously [42], with a few modifications. Briefly, 100 µL of bacterial cells suspended in LB broth (0.5 McFarland) was added to the wells of a sterile flat-bottom 96-well TCP and incubated at 37 °C for 24 h. After the overnight incubation, wells were washed three times with Milli-Q water and air-dried. Next, wells were stained with 100 µL of 0.1% crystal violet (in water) for 30 min. Excess stain was thoroughly removed by three washes with Milli-Q water and then allowed to dry at room temperature for 1 h. Next, 150 µL of 95% ethanol was added to each well for 15 min. As a measurement of biofilm formation, the optical density at 590 nm was read on an ELISA plate reader (FlexStation^®^ 3 Microplate Reader; Molecular Devices). The negative control was LB broth without bacteria and *A. baumannii* (ATCC 19606) was used as a positive control for biofilm production. To quantify the inhibitory effect of melittin peptide on biofilm growth, the method above was employed with some modifications [43]. After overnight incubation, the culture medium was removed and 100 µL of LB broth with melittin (14 µg/mL) was added. After a 2 h incubation at 37 °C, wells were washed, stained, solubilized and the OD was measured at 590 nm. The percentage of biofilm reduction was calculated using the results from bacteria cultured in the absence of peptide. Experiments were performed three times in triplicate, and the data were averaged.

### 2.5. Membrane Permeability Assay

Cytoplasmic membrane damage was determined using steady-state fluorescence, as described before [44], with some modifications. Briefly, *A. baumannii* strains and the *A. baumannii* (ATCC 19606) reference strain were cultivated in LB medium for 24 h. Bacterial cultures were adjusted to approximately 1 × 10^8^ cells/mL in LB broth before treatment with melittin (142 µg/mL) for 2 h at 37 °C. As a positive control, representative cultures of each strain were heat-treated in a water bath at 65 °C for 15 min while untreated bacteria were used as a negative control. Next, cultures were incubated with 30 µM of propidium iodide (PI) at 37 °C for 15 min in the dark. Cells were collected and washed three times in PBS by centrifugation (4000× *g* for 15 min). A final cell suspension was smeared onto a glass slide for imaging on an Axio Imager M2 microscope (Carl Zeiss). Both fluorescence and differential interference contrast (DIC) images were captured for each field of view from multiple areas for the analysis of each treatment group. The DIC image was used for bacteria segmentation and the percentage of PI-positive bacteria was evaluated using Knime workflow. The results from two independent cultures are reported.

### 2.6. Analysis of Bacterial Proliferation

*A. baumannii* strains were cultivated in LB medium for 24 h and cell concentration was adjusted as described above. Then, cells were incubated with 20 µM of carboxyfluorescein succinimidyl ester (CFSE) for 20 min and centrifuged for 5 min (4000× *g*) to remove culture supernatant. After incubation with CFSE, cells were treated with melittin (142 µg/mL) for 2 h at 37 °C. Bacteria were collected and washed three times in PBS by centrifugation (4000× *g* for 15 min). The bacterial suspension was analyzed in an Axio Imager M2 microscope (Carl Zeiss).

### 2.7. Statistical Analysis

Statistical analysis and graphics were performed using R (version 3.6.0) and R Studio. The statistical difference was considered if *p* < 0.05 using a *t*-test and one-way ANOVA.

## 3. Results

### 3.1. Determination of MIC for Melittin, Polymyxin and Imipenem in A. baumannii Strains

*A. baumannii* 100 showed resistance for both polymyxin and Imipenem with a MIC value of 8 and 32 µg/mL, respectively. *A. baumannii* 33677 and 96734 were considered XDR with high MIC only for imipenem (16 µg/mL). The distribution of MIC for polymyxin and imipenem for *A. baumannii* strains are summarized in Table 1.

### 3.2. Biofilm Formation Test

The MIC values of melittin determined for all *A. baumannii* strains showed good activity, with MIC values ranging between 17 and 45.5 µg/mL, except for the PDR strain 100, that maintained viability even in higher concentrations of melittin (284 µg/mL) (Table 1). After the addition of resazurin, a well-known indicator dye for the assessment of viability in both microbial and cell culture applications [45], the fluorescence analysis revealed that 90% of cells were killed at MIC values for all strains, except PDR strain.

After the biofilm formation and staining processes, absorbance analysis at 580 nm revealed that *A. baumannii* ATCC 19606 (the reference strain), 33677 and 96734 formed a weak biofilm, whereas susceptible (31852) and PDR (100) strains formed a moderate biofilm. Despite the observation that susceptible strain 31852 produced a more significant biofilm, reaching approximately 2-fold increase compared to 33677, 96734 and ATCC strains, the melittin activity at sub-inhibitory concentrations (14 µg/mL) affected this strain to a high degree. Melittin treatment significantly reduced its formation of bacterial sessile aggregate by approximately 30%. Melittin did not affect other strains at low tested concentrations (Figure 1).

Analysis of PI staining, a membrane permeability indicator, revealed that bacterial membranes of strains from ATCC were strongly affected by melittin along with pan-resistant strain 100 (Figure 2). Melittin effects on bacterial membrane were evidenced significantly in ATCC, 100 and 31852 strains, as the treatment with this peptide increased in media 13-, 6- and 48-fold membrane disruption, respectively, compared to untreated control. In the reference strain ATCC, melittin produced more damage to the bacteria than the heat-treated positive control. As expected, positive control values maintained a high number of PI fluorescent cells, ranging from 39% to 53%.

To evaluate if melittin exerts a bacteriostatic effect on the *A. baumannii* strains, bacterial cells were stained with CFSE, a cell-permeable fluorescent dye used to monitor cell division (Figure 3). Fluorescence was low in untreated cells of the reference strain from ATCC and merged images showed a high number of unlabeled bacteria, indicating cell division (Figure 3A). In contrast, melittin treatment inhibited the proliferation of this strain, which retained a higher fluorescent signal in the bacterial cytoplasm from an absence of dye dilution during cell division (Figure 3B). Nearly all bacteria observed by DIC showed fluorescence. Untreated bacteria of the PDR strain showed few fluorescent cells as expected (Figure 3C). Melittin treatment did not appear to cause any discernable bacteriostatic effect with few cells still emitting fluorescence as the majority of bacteria were not stained, which correlates to a high proliferation rate (Figure 3D).

## 4. Discussion

Extensive exposure to antibiotics has rapidly increased the propagation of MDR, XDR and PDR bacteria, often dubbed superbugs, which complicates the choice of chemotherapeutics and limits treatment options [46,47]. The increase in antibiotic resistance during biofilm infections is a substantial problem in public health and underlies the need for new, effective solutions. In terms of nosocomial infections, morbidity and mortality due to MDR biofilm-producing *A. baumannii* are of great concern [18,48]. This problem is directly associated with the ability of bacteria to survive and endure in the patient’s body or hospital environment due to biofilm layer production, which is driven by several of yet to be defined molecular mechanisms that lower the diffusion of antibiotics and increase antimicrobial recalcitrance [1,39].

Several strategies have been proposed over the years in an attempt to efficiently treat bacterial biofilms, including prevention, weakening, disruption or killing [49]. Among the limited numbers of new antimicrobials in the pipeline, natural peptides from animal venoms have been demonstrated to possess promising biological properties, which warrant their development as efficacious agents against recalcitrant pathogens [50,51,52]. Among them, melittin from bee venom has been proven to have potent antibacterial activity [31]. There are multiple lines of evidence and several studies that confirm the antibacterial activity of melittin toward antibiotic-resistant bacteria [31,53,54,55,56,57].

Therefore, our study focused on the evaluation of melittin for its in vitro microbicide activity, antibiofilm activity and membrane damage against *A. baumannii* strains isolated from primary infections from hospitals in Brazil with different antibiotic sensitivity profiles. The MICs of melittin, polymyxin, and imipenem against *A. baumannii* strains were compared. According to the results, one isolate was susceptible to imipenem and polymyxin, two were resistant to imipenem and susceptible to polymyxin and one was resistant to both. As a control, susceptibility of *A. baumannii* ATCC 19606 was monitored in this study and the results agreed with previous studies [10,58]. The MICs for melittin ranged from 17 to 45.5 µg/mL, except for PDR that was resistant up to 284 µg/mL. A recent survey reported the synergistic antibacterial, biofilm inhibition and biofilm removal activities of melittin in combination with several antibiotics against MDR, a strong biofilm producer *A. baumannii* strains from clinical isolates. Significant synergistic behaviors were observed combining melittin with colistin and imipenem [57], corroborating previous findings reported by Giacometti et al. in 2003 [59].

The sub-inhibitory concentration of melittin only inhibited biofilm formation of the *A. baumannii* susceptible strain (31852). However, melittin is very effective against biofilm-producing *P. aeruginosa* clinical isolates, with a minimum biofilm inhibition concentration (MBIC) range of 4–16 µM, which was far more active compared to certain antibiotics including ampicillin, chloramphenicol and levofloxacin [56]. Moreover, melittin has been reported to inhibit either biofilm formation or bacterial surface attachment in a time-dependent manner [31]. The peptide was also capable of inhibiting five-strong biofilm-producer strains of MDR *A. baumannii* and inhibiting their biofilm formations, alone or in combination with colistin and imipenem [57]. Noticeably, melittin lessened both biofilm biomass and the viability of biofilm-embedded *Borrelia burgdorferi* strain B31 at different concentrations in comparison to PBS-treated biofilms, which was further confirmed by SYBR Green I/(PI) assay and atomic force microscopy [60]. Another study reported that melittin inhibited biofilm production and destroyed bacterial biofilms [53]. A recent survey implied that melittin was able to penetrate biofilm layers of *P. aeruginosa* gradually and to kill biofilm-residing bacteria kinetically by disrupting the bacterial membrane [61]. Collectively, these shreds of evidence suggest that melittin can decrease biofilm formation, biofilm biomass, and the viability of bacteria within biofilms in a time- and concentration-dependent manner.

The mechanisms of action of antimicrobial peptides against bacteria are diverse. AMPs can act in membrane permeabilization, intracellular targets, and modulation of the immune response [62]. Several lines of evidence suggest that the main mechanism of action of melittin is the formation of toroidal pores in the bacterial membrane [63,64]. Resistant strains were less susceptible to melittin, possibly due to surface modifications in the outer membrane, such as charge modification in lipopolysaccharides driven by phosphorylation, sugar and lipid substitutions [65].

Lipopolysaccharides play a major role in bacterial resistance. Polymyxin B resistance is driven by modifications of lipid A, which decrease electrostatic attraction between the peptide and bacterial membrane [66]. Other mechanisms of resistance against AMPs include proteolytic cleavage, efflux of AMPs that act on intracellular targets and entrapment by matrix proteins and polysaccharides that block AMPs or cause electrostatic repulsion of cationic peptides [65]. Melittin produced a small damage in the bacterial membrane even in a PDR strain subpopulation, but it was not sufficient to efficiently kill this strain. Increased membrane permeability is promising for a drug combination approach against PDR or to avoid resistance. The co-administration of antibiotics with permeability enhancing compounds can enhance activity due to an increase of drug with permeability [67]. Several investigations have addressed possible synergistic effects between melittin and other anti-microbial agents, in particular, conventional antibiotics [55,57,59,68,69].

## 5. Conclusions

The rapid rise of bacterial resistance to currently available antibiotics is one of the major threats to the human population, especially in hospital settings. The development of new effective antibiotics is urgently needed, and peptides are a promising source of microbicidal agents. AMPs such as melittin from bee venom are an interesting alternative for killing resistant strains and to stop their biofilm formation as a remedy for de-contaminating fomites. Evaluation of melittin against Brazilian clinical strains revealed that most strains were susceptible, except for the PDR 100 strain. An analysis of the mechanisms of action suggested that melittin altered the permeability of the plasma membrane, even in a subpopulation of PDR. However, the extent of damage detected in the PDR strain was not sufficient to retard bacterial growth. Collectively, the results further demonstrate that the emergence of “superbugs” and the importance of a continued search for alternative molecules to provide effective remedies. Melittin displayed promising activity against XDR, which suggests that modifications to the peptide sequence could enhance its activity against antibiotic-resistant bacteria to address their threat to public health. Without immediate and global action, the world population is headed for a dangerous post-antibiotic era [33]. Melittin is a highly potent antibacterial agent that may have a good synergistic effect on killing the bacteria and also inhibiting biofilm formation [53].

## Figures and Tables

**Figure 1 antibiotics-09-00178-f001:**
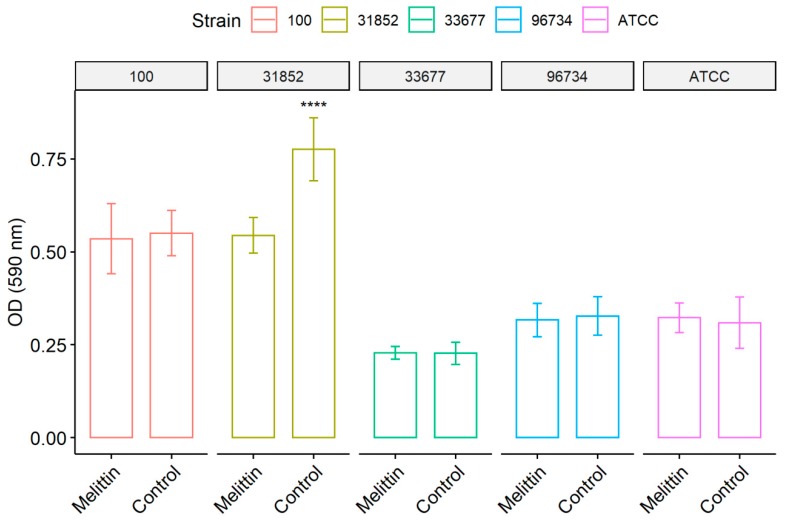
The action of melittin on biofilms of *A. baumannii* strains. Bacteria were allowed to grow in 96-well plates; after 24 h, the biofilms were treated with melittin (142 µg/mL) and after 2 h the biofilms were quantified by staining with crystal violet. Control represents untreated groups. Results represent the mean and standard deviation of at least three independent experiments. **** Statistically significant (*p* < 0.001) using *t*-test.

**Figure 2 antibiotics-09-00178-f002:**
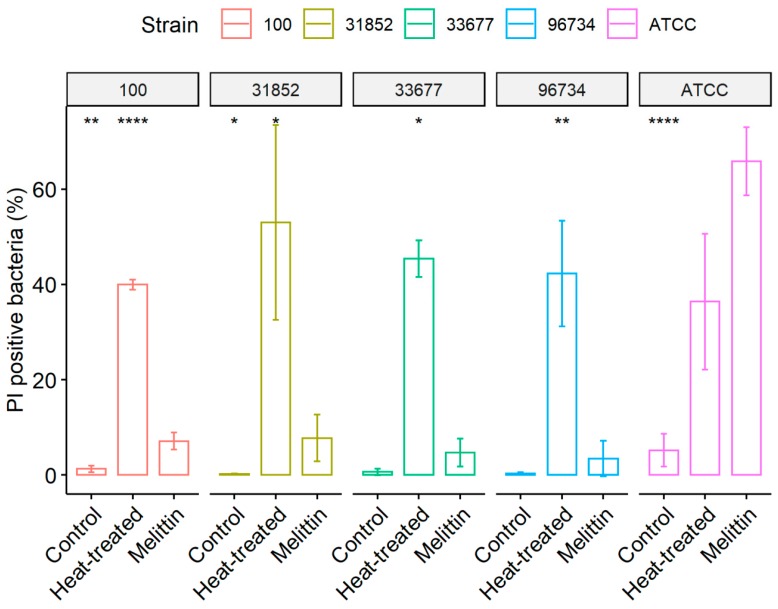
Effect of melittin in bacterial membrane permeability. Different *A. baumannii* strains were treated with melittin (142 µg/mL) for 2 h or heat-treated at 65 °C for 15 min, then incubated with the nucleic acid probe propidium iodide (PI; 30 µM), as a membrane permeability indicator. Bars indicate the percentage of PI fluorescent bacteria in the untreated group, treated with melittin or heat-treated. Results represent the mean and standard deviation. Statistically different (* *p* < 0.05, ** *p* < 0.01 and **** *p* < 0.001) from melittin group using *t*-test.

**Figure 3 antibiotics-09-00178-f003:**
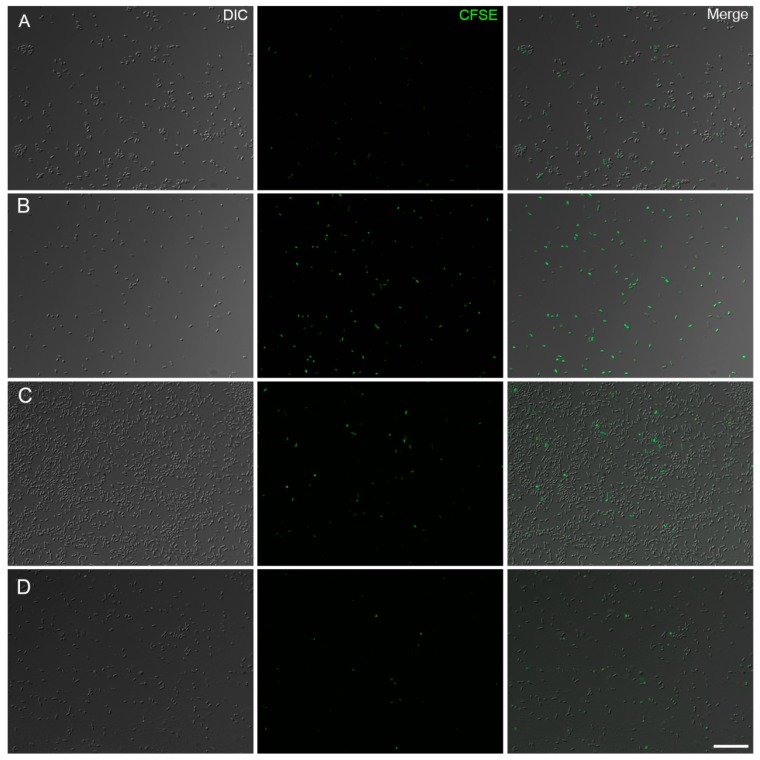
Fluorescence images of CFSE stained *A. baumannii* after treatment with melittin (142 µg/mL) for 2 h at 37 °C. The proliferation of untreated bacteria (**A**) and bacteriostatic effect of melittin against ATCC strain (**B**). *A. baumannii* PDR strain 100 images of CFSE-labeled bacteria from untreated (**C**) and melittin treated cells (**D**). DIC: Differential interference contrast. Bar = 20 μm.

**Table 1 antibiotics-09-00178-t001:** Measured minimum inhibitory concentration (MIC) values for melittin, polymyxin and imipenem.

Strain/Resistance Profile	Melittin (µg/mL)	Polymyxin ^c^ (µg/mL)	Imipenem ^d^ (µg/mL)
*A. baumannii* ATCC 19606	17	0.25	0.25
*A. baumannii* 31852 (S)	20	0.25	≤0.125
*A. baumannii* 33677 (XDR)	31	0.25	16
*A. baumannii* 96734 (XDR)	45.5	0.25	16
*A. baumannii* 100 (PDR ^b^)	>284	8	32

The MIC values were determined using a standard microdilution assay with triplicate samples for each peptide concentration. Identical results were obtained from two separate experiments; therefore, no errors are reported. ^b^ For this strain, melittin did not achieve full killing at the maximum concentration of 85 µg/mL, and in this case, other concentrations with no effect were tested (142 and 284 µg/mL); ^c^ MIC values for polymyxin: S ≤ 2 µg/mL, R ≥ 4 µg/mL; ^d^ MIC values for imipenem: S ≤ 2 µg/mL, I = 4 µg/mL, R ≥ 8 µg/mL.

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
