# Peer review of "Pan-Drug Resistant *Acinetobacter baumannii*, but Not Other Strains, Are Resistant to the Bee Venom Peptide Melittin"

_antibiotics, 2020, doi:10.3390/antibiotics9040178_

Round 1
Reviewer 1 Report
Review of Antibiotics-649418
This article describes the antibiotic effects of the bee venom peptide mellitin against various strains of the pathogenic gram-negative bacterium Acinetobacter baumanii. It contains some very interesting results and for the most part has been carefully prepared. I suggest it is suitable for publication after some minor revisions.
General comments
This article does not contain many display items (two figures and a small table). I am fine with this as long as the journal/editor considers it appropriate This article would benefit from some of the introductory material on mellitin being moved from the Discussion to the Introduction, for example some mention that mellitin acts on lipid bilayers should be made, and at least a few references should be cited to give the reader an idea of how mellitin has previously been investigated as an antibiotic I think that the most interesting result shown is that the resistance of the PDR strain to mellitin. While this result is clearly presented and is dealt with better in the Discussion and Conclusions section, the abstract and title do not make much of this finding. For example, lines 30–36 of the abstract say"the determined minimum inhibitory concentration (MIC) showed that S and XDR strains were susceptible to melittin. In contrast, PDR A. baumannii was resistant to all treatments. Melittin did prevent biofilm formation in a susceptible strain and caused permanent membrane damage in bacteria. Membrane damage was also observed in a subpopulation of PDR, however, it was not sufficient to stop bacterial growth."
This is followed by
"Our results demonstrate the potential of melittin to be used in the control of bacterial infections and suggests that antimicrobial peptides can serve as the basis for the development of new treatments."
I think they should say something about mutations in the PDR that have presumably arisen to confer resistance to widely used therapeutics also confer resistance to mellitin. In fact if it was my article I would title it something like "Pan-drug resistant Acinetobacter baumanni, but not other strains, are resistant to the bee venom peptide mellitin"
Specific comments
Line 21: Need a space after Abstract: and unbold genus name Line 29 and 28–48, and elsewhere in document: Italicised species names sometimes but not always appear very large. Please format consistently. Line 53: Space after delivery Line 95: Maybe add a note here indicating the link of this gene to resistance Throughout document, e.g. lines 111, 112: Add space between number and units, e.g. 24 h Line 141: Is this right 30 M PI? Line 142: "Cells suspension" should be "cell suspension" Table 1: It is a bit hard to compare the MICs between melittin and polymyxin and imipenem since they are in different units. Could you show them all in mol/L? or in both mol/L and g/L? Line 177: add brief note of what resazurin does here to make this easier to read Line 199: replace "Interesting to notice that" with "Interestingly," Line 201: more "sensitive" rather than "sensible" to heat treatment Line 211: "others" should be "other" Line 213: "that could" should be "that it could" Line 260: some formatting problem in the concentration here Line 276: by using "shreds of evidence" are the authors trying to indicate multiple lines of evidence indicate this or that the evidence is scanty? Please reword to clarify Line 304: "and suggests" should be "which suggests" Line 305: "antibiotitc" Line 331 and elsewhere: please capitalise article names in a consistent manner Line 508: please format this reference consistent with othersAuthor Response
Reviewer #1 (Remarks to the Author):
This article describes the antibiotic effects of the bee venom peptide mellitin against various strains of the pathogenic gram-negative bacterium Acinetobacter baumanii. It contains some very interesting results and for the most part has been carefully prepared. I suggest it is suitable for publication after some minor revisions.
Thank you. All major changes in response to the reviewer are written below and highlighted in blue in the manuscript. Minor changes related to the use of English are not denoted. Some lines have changed because of changes suggested by reviewers.
General comments
This article does not contain many display items (two figures and a small table). I am fine with this as long as the journal/editor considers it appropriate This article would benefit from some of the introductory material on mellitin being moved from the Discussion to the Introduction, for example some mention that mellitin acts on lipid bilayers should be made, and at least a few references should be cited to give the reader an idea of how mellitin has previously been investigated as an antibiotic.
We have now moved some of the introductory material on mellitin from discussion to the introduction (Lines 108-118)
I think that the most interesting result shown is that the resistance of the PDR strain to mellitin. While this result is clearly presented and is dealt with better in the Discussion and Conclusions section, the abstract and title do not make much of this finding. For example, lines 30–36 of the abstract say
"the determined minimum inhibitory concentration (MIC) showed that S and XDR strains were susceptible to melittin. In contrast, PDR A. baumannii was resistant to all treatments. Melittin did prevent biofilm formation in a susceptible strain and caused permanent membrane damage in bacteria. Membrane damage was also observed in a subpopulation of PDR, however, it was not sufficient to stop bacterial growth."
This is followed by
"Our results demonstrate the potential of melittin to be used in the control of bacterial infections and suggests that antimicrobial peptides can serve as the basis for the development of new treatments."
I think they should say something about mutations in the PDR that have presumably arisen to confer resistance to widely used therapeutics also confer resistance to mellitin.
We agree with the reviewer and we include this into the abstract (Lines 36-38)
In fact if it was my article I would title it something like "Pan-drug resistant Acinetobacter baumanni, but not other strains, are resistant to the bee venom peptide mellitin"
Thank you for the suggestion. The title was changed to "Pan-drug resistant Acinetobacter baumanni, but not other strains, are resistant to the bee venom peptide mellitin".
Specific comments
Line 21: Need a space after Abstract: and unbold genus name
This was corrected (Line 22).
Line 29 and 28–48, and elsewhere in document: Italicised species names sometimes but not always appear very large. Please format consistently.
Thank you. We changed all instances to a consistently format.
Line 53: Space after delivery
This was corrected (Line 75)
Line 95: Maybe add a note here indicating the link of this gene to resistance Throughout document, e.g.
We indicate the link of this gene to resistance after the description of ST and CC (Lines 128-130)
Lines 111, 112: Add space between number and units, e.g. 24 h
This was corrected (Lines 145, 146).
Line 141: Is this right 30 M PI?
No, this was corrected to 30 µM (Line 175).
Line 142: "Cells suspension" should be "cell suspension"
This was corrected. Thank you (Lines 176-177).
Table 1: It is a bit hard to compare the MICs between melittin and polymyxin and imipenem since they are in different units. Could you show them all in mol/L? or in both mol/L and g/L?
All instances of micro have been substituted with “µ”.
Line 177: add brief note of what resazurin does here to make this easier to read
Thank you. We include brief note of resazurin (Lines 212-213).
Line 199: replace "Interesting to notice that" with "Interestingly,"
We have removed this sentence.
Line 201: more "sensitive" rather than "sensible" to heat treatment
We have removed this sentence.
Line 211: "others" should be "other"
We have change this sentence.
Line 213: "that could" should be "that it could"
We have change this sentence.
Line 260: some formatting problem in the concentration here
This was corrected (Line 282).
Line 276: by using "shreds of evidence" are the authors trying to indicate multiple lines of evidence indicate this or that the evidence is scanty? Please reword to clarify
We reword to “Several lines of evidence suggest that the main mechanism of action of melittin is the formation of toroidal pores in the bacterial membrane” (Lines 306-307).
Line 304: "and suggests" should be "which suggests"
This was corrected (Line 332).
Line 305: "antibiotitc"
This was corrected (Line 333).
Line 331 and elsewhere: please capitalise article names in a consistent manner
Thank you. The references were placed in the same format (Line 358 and elsewhere).
Line 508: please format this reference consistent with others
Thank you. The reference was formatted consistent with others (Lines 542-544).
Submission Date
06 November 2019
Date of this review
18 Nov 2019 10:56:51

Reviewer 2 Report
This manuscript by Rangel et al. is well written and easily readable, though concise, and also presents a good review of the potential use of antimicrobial peptides, and bee venom melittin in particular, against the troublesome bacterial pathogen, A. baumannii (please be consistent and use the correct spelling of the species name!).
However, it presents no novel insights into any aspect of the mechanism of action or potential use of melittin as an antibiotic, and essentially repeats many already published in vitro studies of the efficacy of melittin as an antibiotic against A. baumannii. Some novelty comes in using local clinical isolates, but why these are special in comparison with those used in published studies is not clearly described, and they are poorly characterized. For example, there are no microscopic images to show if there are morphological changes that may be associated with resistance to melittin in the PDR strain.
Specific comments (note these are not exhaustive, but should be addressed in any revision of this manuscript).
Page 2, Abstract: How does this work address “to expand treatment options”? Page 3, line 86: “The activity of melittin against A. baumannii has not been tested to date.” This statement is clearly misleading, since many cited references and other (not-cited) studies have tested this activity, and in a more rigorous way. Page 4, line 110: ug/ml (u=micro) Page 5, line 141: 30 uM propidium iodide (u=micro) Page 5, lines 148-154: Many chromophoric compounds other than DNA will absorb UV light at 260 nm, including metabolites excreted/secreted from healthy cells, so this measure should not be used as a measure of DNA release. Couldn’t you use a more specific method to measure DNA release (e.g., propidium iodide fluorescence). Figure 2B does not present reliable data and should be removed. Specifically: A260 doesn't just measure DNA. Why such high variability among triplicates? Why does DNA release not correlate with PI uptake? Wouldn't it be possible/better to use PI as a DNA stain to measure release into culture supernatants? What is happening at t = 0? Figure 2A: To interpret these data, authors need to show the controls, not just the PI uptake ratios relative to controls, and data from both replicate experiments. What is the effect of heat treatment supposed to be? Evidence that heat treatment leads to membrane permeability needs to be shown (A. baumannii is not a thermophile, but it is resilient to harsh treatments). Page 7, line 201: “sensitive” Page 8, line 212: “led” Page 9, line 217: The statement: “Our results reinforce the evidence of the pore-forming capacity of melittin in baumannii and the promising effect of AMP in rapidly killing bacteria” is not actually supported by the data in a general sense, since neither pore formation nor rapid killing is actually demonstrated; some of the data with the various strains are contradictory. Page 9, line 286-7: How is the statement “mellitin produced a small damage even in PDR strain to the membrane that was not sufficient to lead an outpouring of cellular content” supported by the data?
Author Response
Reviewer #2 (Remarks to the Author):
This manuscript by Rangel et al. is well written and easily readable, though concise, and also presents a good review of the potential use of antimicrobial peptides, and bee venom melittin in particular, against the troublesome bacterial pathogen, A. baumannii (please be consistent and use the correct spelling of the species name!).
However, it presents no novel insights into any aspect of the mechanism of action or potential use of melittin as an antibiotic, and essentially repeats many already published in vitro studies of the efficacy of melittin as an antibiotic against A. baumannii. Some novelty comes in using local clinical isolates, but why these are special in comparison with those used in published studies is not clearly described, and they are poorly characterized. For example, there are no microscopic images to show if there are morphological changes that may be associated with resistance to melittin in the PDR strain.
Thank you. All major changes in response to the reviewer are written below and highlighted in yellow in the manuscript. Minor changes related to the use of English are not denoted. Some lines have changed because of changes suggested by reviewers.
We changed all instances to a consistently format of A. baumannii specie name.
Specific comments (note these are not exhaustive, but should be addressed in any revision of this manuscript).
Page 2, Abstract: How does this work address “to expand treatment options”?
In consideration of the reviewer’s question, we directly state the experiments performed (Lines 27-30) and highlight the impact of the results at the end of the abstract.
Page 3, line 86: “The activity of melittin against A. baumannii has not been tested to date.” This statement is clearly misleading, since many cited references and other (not-cited) studies have tested this activity, and in a more rigorous way.
We have removed this statement.
Page 4, line 110: ug/ml (u=micro)
Page 5, line 141: 30 uM propidium iodide (u=micro)
All instances of micro have been substituted with “µ”.
Page 5, lines 148-154: Many chromophoric compounds other than DNA will absorb UV light at 260 nm, including metabolites excreted/secreted from healthy cells, so this measure should not be used as a measure of DNA release. Couldn’t you use a more specific method to measure DNA release (e.g., propidium iodide fluorescence).
Figure 2B does not present reliable data and should be removed. Specifically: A260 doesn't just measure DNA. Why such high variability among triplicates? Why does DNA release not correlate with PI uptake? Wouldn't it be possible/better to use PI as a DNA stain to measure release into culture supernatants? What is happening at t = 0?
We appreciate the reviewer’s critical comments on the use of A260 as our method for DNA quantification as a measurement of membrane damage. This approach is straightforward and has been utilized by other published works, such as a recent article in Scientific Reports used this same method to measure genetic material release from Pseudomonas aeroginosa (Yasir et al., 2019). However, we agree to remove this data as suggested.
Our working hypothesis for the apparent non-correlation is that the release of nucleic acid from bacteria was probably due to extensive membrane damage caused by melittin treatment. As there are other ways to evaluate cytoplasmic leakage, such as the fluorescent dye PI, we are excluding the OD260 data. We would like to note that the highest variability was mostly observed in the treated groups, probably due to the rapid activity of melittin. Apparently melittin can produce in red blood cells morphological and intracellular alterations with hemoglobin release in seconds after treatment (Hur et al., 2017).
Yasir M, Dutta D, Willcox MDP. Comparative mode of action of the antimicrobial peptide melimine and its derivative Mel4 against Pseudomonas aeruginosa. Sci Rep. 2019 May 8;9(1):7063. doi: 10.1038/s41598-019-42440-2.
Hur J, Kim K, Lee S, Park H, Park Y. Melittin-induced alterations in morphology and deformability of human red blood cells using quantitative phase imaging techniques. Sci Rep. 2017 Aug 24;7(1):9306. doi: 10.1038/s41598-017-08675-7.
Figure 2A: To interpret these data, authors need to show the controls, not just the PI uptake ratios relative to controls, and data from both replicate experiments. What is the effect of heat treatment supposed to be? Evidence that heat treatment leads to membrane permeability needs to be shown (A. baumannii is not a thermophile, but it is resilient to harsh treatments).
The graph was updated to include the untreated control and represent the percentage of PI positive bacteria. Heat treatment was applied as positive control to increase membrane permeability. High temperatures (>60 °C) can produce damage to cell membrane and several articles have demonstrated this event using different techniques (listed below). Also, our results demonstrated that PI uptake was significantly increased in heat treated bacteria.
Ebrahimi A, Csonka LN, Alam MA. Analyzing Thermal Stability of Cell Membrane of Salmonella Using Time-Multiplexed Impedance Sensing. Biophys J. 2018 Feb 6;114(3):609-618. doi: 10.1016/j.bpj.2017.10.032.
Tsuchido T, Katsui N, Takeuchi A, Takano M, Shibasaki I. Destruction of the outer membrane permeability barrier of Escherichia coli by heat treatment. Appl Environ Microbiol. 1985 Aug;50(2):298-303.
Halder, S., Yadav, K.K., Sarkar, R. et al. Alteration of Zeta potential and membrane permeability in bacteria: a study with cationic agents. SpringerPlus 4, 672 (2015) doi:10.1186/s40064-015-1476-7.
Page 7, line 201: “sensitive”
We have removed this sentence.
Page 8, line 212: “led”
We have change the sentence (Lines 259-260)
Page 9, line 217: The statement: “Our results reinforce the evidence of the pore-forming capacity of melittin in A. baumannii and the promising effect of AMP in rapidly killing bacteria” is not actually supported by the data in a general sense, since neither pore formation nor rapid killing is actually demonstrated; some of the data with the various strains are contradictory.
We have removed this sentence.
Page 9, line 286-7: How is the statement “mellitin produced a small damage even in PDR strain to the membrane that was not sufficient to lead an outpouring of cellular content” supported by the data?
We have rewritten this sentence to read:
“Mellitin produced a small damage in bacterial membrane, even in PDR strain subpopulation, but it was not sufficient to efficiently kill this strain.”
Submission Date
06 November 2019
Date of this review
21 Nov 2019 02:40:41

Reviewer 3 Report
This manuscript addresses an important problems involving the drug resistant Acinetobacter baumanii strains in Brazil. It is worthy of study. However, it is highly descriptive. Additionally, it is too small numbers of each strain (S, XDR and PDR) for the study. Unfortunately, I regret inform the authors that your manuscript can not be considered for publication in “Antibiotics”.
Author Response
Reviewer #3 (Remarks to the Author):
Thank you for your comments.
We request that you reevaluate the definition of our sample size as “small”. Within the context of the area of study, our ability to obtain access to clinical samples of bacteria, which represent a grave danger to our healthcare systems, is a major success. Too often, researchers have no choice but to work with a restricted group of representative strains that can become “domesticated” to the lab environment. It is difficult to bridge basic science with clinical science. By publishing our work, your journal has an opportunity to demonstrate that these types of collaborations will be recognized and valued.
As to the characterization of our work as only descriptive, we believe that it undervalues our experimental approach to address our hypothesis that naturally occurring peptides can positively contribute to resolving a healthcare issue that is only getting worse. Are elements description? Yes, but this is the beginning of our studies and we can only continue by securing funding, which requires a demonstration of progress, ie publication.
Reviewer 4 Report
Abstract and Introduction sections:
Please make uniform the characters.
Table 1:
Please express in uniform way the MIC related to Melittin, polymixin and imipenem.
Figure 1:
Please check the level of statistical significance. On the basis of the S.D. it is questionable the P<0.001 level of significance.
Author Response
Reviewer #4 (Remarks to the Author):
Thank you. All major changes in response to the reviewer are written below and highlighted in the manuscript. Minor changes related to the use of English are not denoted. Some lines have changed because of changes suggested by reviewers.
Abstract and Introduction sections:
Please make uniform the characters.
Table 1:
Please express in uniform way the MIC related to Melittin, polymixin and imipenem.
Figure 1:
Thank you. The MICs between melithin and polymyxin and imipenem were placed in the same units µg/ml in the table 1 and throughout the text.
Please check the level of statistical significance. On the basis of the S.D. it is questionable the P<0.001 level of significance.
Statistical analysis was performed using R software, and the level of significance is correct for t-test.
Below is the statistic summary generated by the software.
# A tibble: 1 x 8
.y. group1 group2 p p.adj p.format p.signif method
<chr> <chr> <chr> <dbl> <dbl> <chr> <chr> <chr>
1 OD Melittin Control 0.0000472 0.000047 4.7e-05 **** T-test
Submission Date
06 November 2019
Date of this review

Round 2
Reviewer 2 Report
The authors have changed the title to emphasize their novel insight into the unexpected high-level resistance of a clinical pan-drug-resistant A. baumannii strain to melittin, as recommended by one of the reviewers. They have also removed ambiguous (but not necessarily contradictory) data and added in new experiments as requested by the reviewers. All of the reviewers' concerns have been addressed.
If the authors will have the opportunity to edit the manuscript again, I suggest they further improve formatting and English expression to reflect journal style and standards.
Author Response
Reviewer #2 Review 2:
The authors have changed the title to emphasize their novel insight into the unexpected high-level resistance of a clinical pan-drug-resistant A. baumannii strain to melittin, as recommended by one of the reviewers. They have also removed ambiguous (but not necessarily contradictory) data and added in new experiments as requested by the reviewers. All of the reviewers' concerns have been addressed.
If the authors will have the opportunity to edit the manuscript again, I suggest they further improve formatting and English expression to reflect journal style and standards.
We thank the Reviewer and the helpful suggestions and corrections of the references have improved the quality of the paper. The text was adequate for the "instructions for authors" and some other minor corrections introduced (below). A reference that was incomplete (not yet formally published) has been replaced.
Title- formatted according to the Journal style
Line 13-Laboratorio de Ultraestrutura Celular - Laboratory of Cellular Ultrastructure
Line 15-The sentence of the “author address” was changed
Line 23-Concernigly- Concerning
Line 39-keywords (added) - antimicrobial peptides
Line 75- Concerning -regarding
Line 216-Subinhibitory- sub-inhibitory
Line 223-96-well microtiters plates- 96-well plates
Minute- min in all text
All references have been checked, incomplete characters corrected.
Was inserted- Author contribution

Reviewer 3 Report
This second version of the paper is much improved. I understood what you said.
Author Response
Reviewer #3 Review 2:
This second version of the paper is much improved. I understood what you said.
Thank you. The text was adequate for the "instructions for authors" and some other minor corrections introduced. A reference that was incomplete (not yet formally published) has been replaced.
